# Predictors of Umbilical Venous Catheter Misalignment

Aimann Surak [1,*] , Michael Miller [2] and Henry Roukema [2]

1    Department of Pediatrics, Royal Alexandra Hospital, Edmonton, AB T5H 3V9, Canada
2    Department of Pediatrics, London Health Sciences Centre, London, ON N6A 5W9, Canada
*    Correspondence: aimannsurak@gmail.com

**Abstract:** Introduction: The insertion of an umbilical venous catheter (UVC) is a routine procedure. The success rate of this procedure is about 40–50%, with potential complications arising from misaligned UVC placement. Objectives: To explore potential factors that may aid in the prediction of UVC misalignment. We hypothesized that UVC misalignment is proportionally related with increased chronological age. Methods: Retrospective chart review for newborns who had an UVC procedure followed by an x-ray. All analyses were conducted using standard comparative statistical methods and logistic regression modelling with SPSS v.24 (IBM Corp., Armonk, NY, USA), and *p*-values < 0.05 were considered statistically significant. Results: The final sample size was 480 patients. There were significant differences between the two groups in terms of gestational age {OR 1.06, 95% CI (1.02–1.10)}, small for gestation (SGA) status {OR 1.07, 95% CI (0.98–1.15)}, and 5-min APGAR scores {OR 0.48, 95% CI (0.23–1.00)}. There were no other significant group differences. Logistic regression modeling identified that chronologic age positively predicted, and SGA negatively predicted, UVC misalignment. *Conclusion:* A misaligned UVC is more likely to occur in late preterm and term babies, whereas a baby being SGA increases the likelihood of a well-aligned UVC.

**Keywords:** UVC; ductus venosus; preterm; NICU

## What Is Known

There is an almost 20% chance that the umbilical venous catheter is misaligned in the portal venous system instead of taking the correct track towards the ductus venosus-IVC junction.

## What Is New

Our study confirms that larger-sized babies are more likely to have their UVC misaligned compared to smaller babies.

## 1. Introduction

The insertion of an umbilical venous catheter (UVC) is a routine procedure to obtain vascular access for both premature and term newborns (Figure 1) [1]. Approximately 13–19% of UVCs will not follow the correct path towards the ductus venosus, and instead, will misalign and follow the hepatic or portal veins into one of the hepatic lobes [2,3]. Potential complications of incorrect UVC placement include arrhythmias, intracardiac thrombosis, systemic and pulmonary embolization, endocarditis, myocardial perforation, pericardial effusion, pleural effusion, and pulmonary infarction and hemorrhage [4].

Upon proper alignment in the ductus venosus, the ideal UVC position to minimize complications is in the thoracic inferior vena cava, or at the right atrial-inferior vena cava junction [5,6]. Two of the most common methods used to guide insertion length include the shoulder–umbilicus length graph and a regression equation based on birth weight [7,8]. However, misalignment in the hepatic or portal circulation is common.

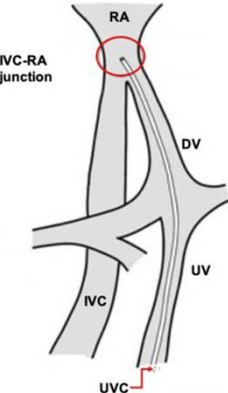

**Figure 1.** Correct umbilical venous catheter position.

Currently, no studies have examined risk factors predicting UVC misalignment. The objective of this study is to explore potential factors that contribute to UVC misalignment, with misalignment being defined as "the catheter following the incorrect path".

We hypothesized that UVC misalignment is directly correlated with advanced chronological age.

## 2. Methods

### 2.1. Study Design

We performed a retrospective cohort study of neonates who underwent umbilical venous catheterization followed by X-ray assessment of positioning using chart review. This was between 1 June 2011 and 30 June 2017, in the neonatal intensive care unit (NICU) at Children's Hospital, London Health Sciences Centre (LHSC) in London, Ontario, Canada. Our institution is a tertiary care center serving a population of approximately 1.5 million people in southwestern Ontario. The study population was identified through the local database; all procedure notes were reviewed accordingly. The study was approved by the Review Ethics Board at Western University.

### 2.2. Protocol

UVC procedures were performed by nurse practitioners, fellows, or residents under supervision. The UVC position was confirmed by both anterio-posterior and lateral view chest/abdominal X-ray. Good UVC alignment was defined when the UVC tip was at the diaphragm level or within 1 cm above or below the diaphragm level on the lateral view, when it was towards the right atrium; otherwise, it was considered misaligned. The X-ray was interpreted by authors AS and HR, both neonatologists with experience with the UVC insertion procedure and x-ray interpretation. X-ray reports by radiologists were also retrospectively reviewed with no blinding to make sure there were no discrepancies in the interpretation.

Inclusion criteria were all newborns at any age in the NICU who had a UVC procedure performed followed by an x-ray. Exclusion criteria consisted of urgent UVC insertions for resuscitation purposes, subsequent UVC attempts if the first attempt resulted in a misaligned UVC, and outborn patients who had a UVC procedure done prior to transfer to our institution, whether it was done by the community physician or by the transport team.

### 2.3. Statistical Methods

With an anticipated UVC misalignment rate of 20%, we anticipated approximately 120 outcomes of interest over the 5-year study period (N = 600). This sample size allowed for many predictors (up to 24) of UVC misalignment with 5% significance, 90% power, and 10% attrition. We collected information for gestational age, birth weight, age of insertion in hours, APGARs at 1 and 5 min, cord gases (venous and arterial), sex, small for gestation (SGA) status, UVC size, and performer (nurse practitioner, resident, fellow or staff).

Descriptive statistics and frequencies were calculated for all continuous and categorical variables, respectively. Two groups were analyzed and compared, the well aligned and misaligned groups, with independent t-tests or Mann–Whitney U tests, when appropriate; chi-square tests were used to compare UVC groups for categorical variables. A multivariable logistic regression model was used to examine chronological age in hours at UVC insertion as a predictor of UVC misalignment, along with secondary predictors (e.g., weight, sex, APGAR, etc.) that were significant at $p < 0.05$ on bivariate analyses. For exploratory purposes, a Cox regression survival analysis was also used to examine the difference in time from birth to UVC insertion between infants with properly positioned and misaligned UVCs. All analyses were conducted with SPSS v.24 (IBM Corp., Armonk, NY, USA), and $p$-values $< 0.05$ were considered statistically significant.

## 3. Results

The final cohort included 480 patients; 332 (~70%) met the definition of good alignment and 148 (~30%) of misalignment (Figure 2). The relationships of chronologic age, small for gestational age (SGA) status, and 5 min APGAR scores to UVC malposition are shown in Table 1. Gestational age, birthweight, 5-min APGAR score, and SGA status were all significant predictors of malposition at the bivariate level ($p < 0.05$). Due to multicollinearity between gestational age and birthweight, birth weight was dropped from further analysis, and gestational age, 5-min APGAR score, and SGA status were included in a final multivariable logistic regression model (see Table 1). Gestational age positively predicted malposition over and above 5-min APGAR score and SGA status; specifically, a 1-week increase in GA increased the likelihood of malposition by 1.06 (95%CI: 1.02–1.10). SGA status was negatively associated with malposition ($p = 0.049$), but the upper end of the 95% CI extended to 1.00, which presents some unreliability in the finding. There was no significant effect of 5-min APGAR scores.

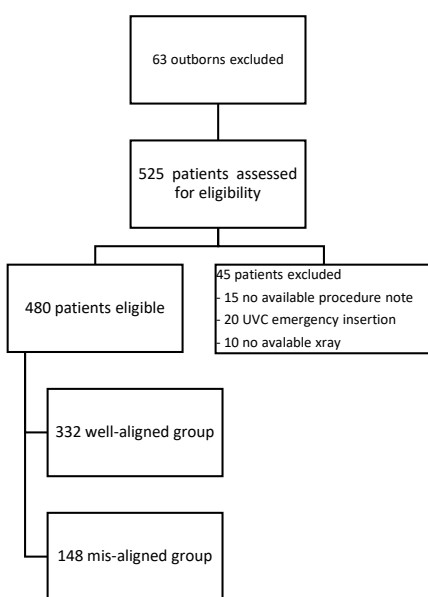

**Figure 2.** Flow diagram for the study population.

**Table 1.** Relation of gestation, SGA and 5 min APGARs to UVC malposition.

| Variable | B | Standard Error | OR (95% CI) | *p*-Value |
|---|---|---|---|---|
| Chronological age | 0.06 | 0.02 | 1.06 (1.02–1.10) | 0.004 |
| APGARs at 5 min | 0.06 | 0.04 | 1.07 (0.98–1.15) | 0.121 |
| SGA | −0.73 | 0.37 | 0.48 (0.23–1.00) | 0.049 |

## 4. Discussion

The conclusion of our study is that it is more likely to have a misaligned UVC in late preterm and term babies, whereas being SGA makes it more likely to have a well-aligned UVC.

Our study found a proportional correlation between chronological age and misalignment of the UVC in late preterm and term babies, whereas being SGA makes it more likely to have a well-aligned UVC. Performing the procedure earlier in the first few hours of life does not increase the chance of success.

It is logical to assume that a UVC line will follow the pathway with higher blood flow. Animal experiments have shown that 50% of the umbilical blood flow was shunted through the ductus venosus, and that the shunted fraction could reach 70% during hypoxemia [9]. Babies who are SGA have more often experienced hypoxemia or hypoperfusion in utero, which may predispose them to higher flow through the ductus venosus [10,11]. This could explain why being SGA makes it more likely to have a well-aligned UVC, especially if they are exposed to hypoxia; we could not establish this relation in our study, most likely due to its retrospective nature, as well as due to the limited available data for hypoxia in these SGA infants. Other limitations of our study include the lack of blinding when interpreting the X-ray findings, as well as the lack of inclusion of procedures performed prior to arrival to our center.

Lind et al. [12] demonstrated the function of the ductus venosus by angiography in previable fetuses, and in 1971, Rudolph et al. [13] showed 55% shunting through the ductus venosus in 33 exteriorized human fetuses of 10 to 20 weeks' gestation.

Kiserud et al. revealed that the average fraction shunted through the ductus venosus to be 28% to 32% at 18 to 20 weeks, decreasing to 22% at 25 weeks, and reaching 18% at 31 weeks; he concluded that in the human fetus, a higher proportion of umbilical blood is directed to the liver and less is shunted through the ductus venosus with increasing gestational age [14,15]; this could explain why chronological age positively predicted UVC misalignment. Studies with blinding methodology and involvement of ultrasound-guided UVC insertions are needed.

**Author Contributions:** A.S. reviewed charts, collected data and drafted the manuscript; M.M. performed the statistical analysis and reviewed the manuscript; H.R. supervised data collection and manuscript. All authors have read and agreed to the published version of the manuscript.

**Funding:** This research received no external funding.

**Institutional Review Board Statement:** Study was approved by research ethics board of Western university, London, Ontario, Canada.

**Informed Consent Statement:** Not applicable.

**Conflicts of Interest:** The authors declare no conflict of interest.

## Abbreviations

NICU    Neonatal intensive care unit
SGA     Small for gestation
UVC     Umbilical venous catheter

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
