# Peer review of "Predictors of Umbilical Venous Catheter Misalignment"

_pediatrrep, doi:10.3390/pediatric14040047_

Round 1

Reviewer 1 Report

This should be good as a brief communication.  It is well presented and I have a few comments.

As noted later, I did not have a "diagram 1"in my file.  However, I was going to suggest that a diagram showing a properly aligned UVC vs an improperly aligned UVC would be helpful.  

Abstract, 1st sentence:  Please state how "success rate" is defined.  40-50% seems very low from my experience.

Section 2.2: Were there any differences (significant or not) in the success rate among these practitioners?

Line 92:  I did not see "diagram 1" in the file I received.

Lines 96-97 and 105-6: Please define what "Performing the procedure earlier" means.  Earlier experience? Earlier gestational age?  The main theme of the report is that earlier GA gave fewer misalignments. 

Author Response

Thank you for your reviews and comments. Please see my point-to-point responses as well as attached revised manuscript. 

  • Could you please correct he second author first name from Mchael to Michael.
  • Please see my point-to-point responses to the reviewers’ comments.

Reviewer 1:

As noted later, I did not have a "diagram 1"in my file.  However, I was going to suggest that a diagram showing a properly aligned UVC vs an improperly aligned UVC would be helpful.  

Please see attached revised manuscript. Diagram and a figure were added accrdingly. 

Abstract, 1st sentence:  Please state how "success rate" is defined.  40-50% seems very low from my experience.

The definition is highlighted in the introduction section (line 36-37-38).

(About 40-50% of UVCs will not follow the correct path towards the ductus venosus and instead, it will misalign and follow the hepatic or portal veins into one of the hepatic lobes.) 

Section 2.2: Were there any differences (significant or not) in the success rate among these practitioners?

As highlighted in the result section (line 96) there was no significant difference.

Line 92:  I did not see "diagram 1" in the file I received.

Please see attached revised manuscript. Diagram was added. 

Lines 96-97 and 105-6: Please define what "Performing the procedure earlier" means.  Earlier experience? Earlier gestational age?  The main theme of the report is that earlier GA gave fewer misalignments. 

I meant by earlier is the postnatal age in hour. Please see attached revised manuscript (revised accordingly) (highlighted in line 97 and line 107-108). 

Reviewer 2 Report

Review: UVC complications for Children

Major comments:

Background:

1.       There is no reference for the 40-50% malpositioned rate. This is quite high with respect to a quick literature search that included the following references:

a.       Incidence <13%: Levit OL et al. Umbilical catheter-associated complications in a level IV neonatal intensive care unit. J Perinatol. 2020 Apr;40(4):573-580.  doi: 10.1038/s41372-019-0579-3.   

b.        Incidence 19.4%: Mutlu M. Umbilical venous catheter complications in newborns: a 6-year single-center experience. J Matern Fetal Neonatal Med. 2016 Sep;29(17):2817-22. doi: 10.3109/14767058.2015.1105952. Epub 2015 Nov 2.

Perhaps include the reference to justify you sample size calculation.

2.       Through the background, and in fact the paper, there is an emphasis on malpositioning into the wrong vessel. However, here and there the authors mention malpositioning that includes incorrect depth. These comments do not seem well integrated into the text. The authors should define malpositioning at the start, and use this definition throughout. If the definition includes depth, then this must be discussed and justified in the background, and the incidence of depth malpositoning addressed in the results, and a comparison to other literature or implications addressed in the discussion. If the definition for this paper does not include depth, then remove all mention of it after provision of the definition.

Methods:

1.       Description of determination of UVC alignment and positioning needs to be more rigorous: Who read the x-rays and interpreted position? Was this someone blinded to the other study data? If not, why not? Ideally all would be read by a radiologist, or at least by someone who was not involved in collecting the other study data. Were both AP and lateral films standard? How was positioning determined if there was no lateral film? Line 64: what does it mean that the UVC was in the “right track”?

2.       Why exclude the UVC insertions by the transport team? They are likely the best at it, and they tend to x-ray prior to leaving.

3.       What data elements did you collect for each patient? You’ve indicated that you could assess up to 24 predictor variables; did you collect that many?

4.       How did you arrive at 24 predictor variables? General rule of thumb of which I am aware is 15 cases with complete data collection for every predictor variable. Given 250 misalignments, 10% attrition so 25 with incomplete data, you could assess 225/15 = 15 variables. But I readily confess that this is not my area of expertise, so best to reference the source you used for this calculation.

Results:

5.       I do not have access to Diagram 1, so do not know to what it refers. I am assuming (hoping) it is a flow diagram detailing inclusions and exclusions.

6.       Please state the number of patients who have normally aligned and misaligned UVCs in the text.

7.       Would like to know the number (%) who were malaligned for vessel, and the number (%) for depth of insertion. As it stands I do not know the incidence of malpositioned UVCs, so cannot provide comment on how this was addressed in the discussion.

Discussion:

8.       Line 104: use of Proportional correlation. Did your modeling demonstrate a linear correlation? Or is this a direct correlation?

9.       I’d propose that there should be a comparison of their incidence of malpositioning with other sources from the literature.

10.   Lines 108-111: 1. Catheter follows high blood flow, 2. Animals (in utero or after birth?) have more than 50% of blood going through the ductus venosus, and more in hypoxia. 3. SGA must therefore have higher success rate. You need an argument to connect these dots. Are patients who are SGA necessarily hypoxic? Won’t this depend on whether it is in utero or after birth?

11.   Lines 113-115 seem unnecessary.

12.   The discussion only addresses malalignment into the incorrect vessel. However, if the UVC goes too high, or is just sitting too low, this is malaligned. This would not be predicted by poor blood flow.

13.   The authors mention that it is logical for the catheter to go where the most blood does. Ideally this should be referenced and there should be literature cited to support this supposition.

14.   Does the literature that addresses the complications of UVC comment on the relative burden of misaligned catheters? The conclusions should not include discussion of the adverse effects, as these were not the subject of study in this manuscript.

15.   Discussion should have a (brief) mention of strengths and weaknesses

16.   Line 122: The authors have not provided evidence for “increasing literature” addressing complications of malpositioning. In fact, it is relatively rare to have complications of malpositioning.

Minor comments:

Overall:

1.       Multiple sentences throughout are missing an article, use inappropriate prepositions, or have commas placed incorrectly. Would benefit throughout from thorough grammatical editing.

2.       One of the author’s first names have been entered incorrectly: Should read Michael Miller.

Abstract:

3.       Methods. Too much detail about exclusion criteria for an abstract.

4.       Methods: How did you compare the groups?

5.       Define SGA at first use in abstract

Background:

6.       Need references in Lines 35-36. Lines 36-38.

7.       Line 44 – “based on these studies”. What studies? The reader cannot assume you are referring to a particular reference, as there are several in the line before.

8.       Line 47: replace “trials” with “studies”.

9.       Line 50: Did you actually mean “proportionally correlated” or just “directly correlated”? Proportionality would imply that the ratio of “UVC misalignment incidence” and “age” are a constant ratio.

10.   Line 50: You mention using chronological age, but UVCs are rarely placed beyond the first 1-2 days of life. Is this supposed to be gestational age? I note that you performed an exploratory analysis (in the methods) of the relationship between time from birth (chronological age) and misalignment, though an exploratory section of your analysis can hardly form the primary element of your hypothesis.

Methods

11.   I would suggest referring to “gestational age” or “corrected gestational age” throughout, using the relevant abbreviation, and then being specific about chronological age when it is relevant (e.g. line 80 needs more specificity).

Results:

12.   Lines 89-92: Repetition of the methods. Remove.

13.   Line 95: “with significant p values”. This is redundant; you have already defined the p-value at which you consider significance, and you have already stated that there is a significant difference. Remove.

14.   Line 96 – “Performing the procedure earlier” is ambiguous. Does this mean earlier in life? Earlier by gestational age?

15.   Define SGA at the first use.

Discussion:

16.   Lines 106-107: This sentence needs entirely reworded to be suitable for academic writing.

17.   Line 108 – remove first sentence

Author Response

Thank you for your reviews and comments. Please see my point-to-point responses as well as attached revised manuscript. 

Reviewer 2:

Background:

  1. There is no reference for the 40-50% malpositioned rate. This is quite high with respect to a quick literature search that included the following references:

References were changes accordingly. Please see highlighted modified mal-positioned rate.

  1. Through the background, and in fact the paper, there is an emphasis on malpositioning into the wrong vessel. However, here and there the authors mention malpositioning that includes incorrect depth. These comments do not seem well integrated into the text. The authors should define malpositioning at the start, and use this definition throughout. If the definition includes depth, then this must be discussed and justified in the background, and the incidence of depth malpositoning addressed in the results, and a comparison to other literature or implications addressed in the discussion. If the definition for this paper does not include depth, then remove all mention of it after provision of the definition.

Definition was revised and highlighted (line 36-37-38), also the concept of “depth” was removed.

Methods:

  1. Description of determination of UVC alignment and positioning needs to be more rigorous: Who read the x-rays and interpreted position? Was this someone blinded to the other study data? If not, why not? Ideally all would be read by a radiologist, or at least by someone who was not involved in collecting the other study data. Were both AP and lateral films standard? How was positioning determined if there was no lateral film? Line 64: what does it mean that the UVC was in the “right track”?

Please see highlighted sections in the methods section. For the “right track” wording, it was removed to avoid confusion.

  1. Why exclude the UVC insertions by the transport team? They are likely the best at it, and they tend to x-ray prior to leaving.

This was more related to the availability of the charts and the procedure notes done by the transport team. It is mainly for logistic purposes.

  1. What data elements did you collect for each patient? You’ve indicated that you could assess up to 24 predictor variables; did you collect that many?

We didn’t collect 24 elements. We meant that we could collect up to 24 elements. However, we collected gestational age, birth weight, age of insertion in hours, APGARs at 1 and 5 minutes, cord gases (venous and arterial), sex, SGA status, UVC size, performer (nurse practitioner, resident, fellow or staff).

Please see revised methods (highlighted).

  1. How did you arrive at 24 predictor variables? General rule of thumb of which I am aware is 15 cases with complete data collection for every predictor variable. Given 250 misalignments, 10% attrition so 25 with incomplete data, you could assess 225/15 = 15 variables. But I readily confess that this is not my area of expertise, so best to reference the source you used for this calculation.

Please see response to previous comment.

Results:

  1. I do not have access to Diagram 1, so do not know to what it refers. I am assuming (hoping) it is a flow diagram detailing inclusions and exclusions.

Attached in the revised manuscript.

  1. Please state the number of patients who have normally aligned and misaligned UVCs in the text.

Stated and highlighted in the text.

  1. Would like to know the number (%) who were malaligned for vessel, and the number (%) for depth of insertion. As it stands I do not know the incidence of malpositioned UVCs, so cannot provide comment on how this was addressed in the discussion.

We didn’t collect that data. The misalignment group included both the misalignment for vessel and misalignment for depth. 

Discussion:

  1. Line 104: use of Proportional correlation. Did your modeling demonstrate a linear correlation? Or is this a direct correlation?

It was only a direct correlation.

  1. I’d propose that there should be a comparison of their incidence of malpositioning with other sources from the literature.

Thank you for suggestion. We found no other similar studies in literature for comparison. Most of the studies looked at the malposition in terms of depth but not the misalignment in the vessel.

  1. Lines 108-111: 1. Catheter follows high blood flow, 2. Animals (in utero or after birth?) have more than 50% of blood going through the ductus venosus, and more in hypoxia. 3. SGA must therefore have higher success rate. You need an argument to connect these dots. Are patients who are SGA necessarily hypoxic? Won’t this depend on whether it is in utero or after birth?

Absolutely. I added this paragraph:

“especially if they are exposed to hypoxia; we could not establish this relation in our study, most likely due to the retrospective nature of it as well as limited available data for hypoxia in these SGA infants.”

  1. Lines 113-115 seem unnecessary.

Those lines were added to explain the percentage of blood flow. But I am happy to remove if they seem redundant or unnecessary.

  1. The discussion only addresses malalignment into the incorrect vessel. However, if the UVC goes too high, or is just sitting too low, this is malaligned. This would not be predicted by poor blood flow.

Agree. However, our study focused on misalignment only in the vessel not depth as mentioned above.

  1. The authors mention that it is logical for the catheter to go where the most blood does. Ideally this should be referenced and there should be literature cited to support this supposition.

Agree; I tried to cite a reference from the literature, but I couldn’t find such study unfortunately.   

  1. Does the literature that addresses the complications of UVC comment on the relative burden of misaligned catheters? The conclusions should not include discussion of the adverse effects, as these were not the subject of study in this manuscript.

Yes, please see highlighted lines 38-41 in the introduction.

  1. Discussion should have a (brief) mention of strengths and weaknesses

Please see highlighted lines 123-127.

  1. Line 122: The authors have not provided evidence for “increasing literature” addressing complications of malpositioning. In fact, it is relatively rare to have complications of malpositioning.

Those lines were removed to also reflect your suggestion to not include side effects and complication in this section.

Minor comments:

Overall:

  1. Multiple sentences throughout are missing an article, use inappropriate prepositions, or have commas placed incorrectly. Would benefit throughout from thorough grammatical editing.

I modified accordingly.

  1. One of the author’s first names have been entered incorrectly: Should read Michael Miller.

Corrected.

Abstract:

  1. Too much detail about exclusion criteria for an abstract.

Modified accordingly and highlighted.

  1. Methods: How did you compare the groups?

Modified accordingly and highlighted.

  1. Define SGA at first use in abstract

Defined.

Background:

  1. Need references in Lines 35-36. Lines 36-38.

References added.

  1. Line 44 – “based on these studies”. What studies? The reader cannot assume you are referring to a particular reference, as there are several in the line before.

Removed.

  1. Line 47: replace “trials” with “studies”.

Replaced.

  1. Line 50: Did you actually mean “proportionally correlated” or just “directly correlated”? Proportionality would imply that the ratio of “UVC misalignment incidence” and “age” are a constant ratio.

We meant directly correlated. Modified accordingly.

  1. Line 50: You mention using chronological age, but UVCs are rarely placed beyond the first 1-2 days of life. Is this supposed to be gestational age? I note that you performed an exploratory analysis (in the methods) of the relationship between time from birth (chronological age) and misalignment, though an exploratory section of your analysis can hardly form the primary element of your hypothesis.

We meant the chronological age. Agree that the procedure is rarely performed beyond the 1-2 days of life. Please see added table one clarifying the analysis.

Methods

  1. I would suggest referring to “gestational age” or “corrected gestational age” throughout, using the relevant abbreviation, and then being specific about chronological age when it is relevant (e.g. line 80 needs more specificity).

Modified accordingly.

Results:

  1. Lines 89-92: Repetition of the methods. Remove.

Removed.

  1. Line 95: “with significant p values”. This is redundant; you have already defined the p-value at which you consider significance, and you have already stated that there is a significant difference. Remove.

Removed.

  1. Line 96 – “Performing the procedure earlier” is ambiguous. Does this mean earlier in life? Earlier by gestational age?

We meant earlier in life, like in the first few hours versus on day 2 or 3. Highlighted and clarified.

  1. Define SGA at the first use.

Defined.

Discussion:

  1. Lines 106-107: This sentence needs entirely reworded to be suitable for academic writing.

Reworded.

  1. Line 108 – remove first sentence

Removed.

Round 2

Reviewer 2 Report

Review: UVC complications for Children, 2

Minor comments:

Overall:

1.       Multiple sentences throughout are missing an article, use inappropriate prepositions, or have commas placed incorrectly. Would benefit throughout from thorough grammatical editing. – this has not been addressed with the first revision. .

2.       There remains quite a bit of redundancy in wording. For example:

“Logistic regression model revealed that chronological age positively predicted UVC misalignment, whereas SGA negatively predicted UVC malpositioning” could be shortened with no change in meaning to: Logistic regression modeling identified that chronologic age positively predicted, and SGA negatively predicted, UVC misalignment”.

This is just one example. Excess word use is present throughout the manuscript.

3.       Multiple sentences are incomplete – e.g. they do not start with an appropriate article. This may be acceptable in abstracts, but not in the body of the manuscript.

Abstract:

4.       Methods: Lines 6-7, by saying that you did a retrospective chart review, you have described your methods of carrying out your methodology. It is arguably more important to state your methodology. In this case, your methodology is retrospective cohort study, where your cohort is children who had a UVC placed followed by x-ray. Your methods are chart review. You could say: “We performed a retrospective cohort study of neonates who underwent umbilical venous catheterization followed by x-ray assessment of positioning using chart review.”

5.       Results: Lines 10-11 the authors begin talking about “the two groups” without ever explaining what the two groups will be or who is being compared. I’d suggest at the beginning of the results, report the number of patients with UV catheters (the cohort) and then also reporting the number (%) that were misaligned, which will make it easy for the reader to identify that this is a subgroup of the cohort, and then it is reasonable to assume that the rest are the other group.

6.       Results: Lines 11-13. Reporting the results of a logistic regression would usually involve reporting a quantitative amount of change (95% CI) for each unit change in the relevant predictor variable. There needs to be some numeric demonstration of this in the results, as it is the primary outcome. This will justify statements of positive and negative prediction.

Background:

1.       I like the figure!

2.       Line 44: Either remove “following the incorrect path”, or change the sentence to “…is to explore potential factors that contribute to the UVC misalignment, with misalignment defined as the catheter following the incorrect path”.

Results:

7.       Line 89: This sentence needs reworded. Suggest: The final cohort included 480 patients; 332 met the definition of good alignment and 148 of misalignment (Diagram 1)

8.       Line 90: Should read “The relationships of chronologic age, small for gestational age (etc)….to UVC malpositioning are shown in Table 1.”

9.       Lines 92-95: This is awkward reporting. Normally would be reported along the lines of the following, which is suggested by Zach Bobbitt from Statology (https://www.statology.org/how-to-report-logistic-regression-results/#:~:text=We%20can%20use%20the%20following,%5D%20and%20%5Bresponse%20variable%5D.):

Logistic regression was used to analyze the relationship between [predictor variable 1], [predictor variable 2], … [predictor variable n] and malposition.

It was found that, holding all other predictor variables constant, the odds of malposition occurring [increased or decreased] by [some percent] (95% CI [Lower Limit, Upper Limit]) for a one -unit increase in chronological age.

It was found that, holding all other predictor variables constant, the odds of malposition occurring [increased or decreased] by [some percent] (95% CI [Lower Limit, Upper Limit]) for a one -unit increase in birthweight.

10.   Line 94: how do you justify that SGA is the overall best predictor?

11.   Line 96-97: I found this confusing. On the one hand, the authors have indicated that increasing chronologic age in hours predicts misalignment. But this sentence says there is no difference in the first few hours versus the 2nd or 3rd days. Needs more consistency.

Discussion:

12.   Lines 111-113: There is still no connection made between how hypoxemia, which increases ductus venosus (DV) flow, would explain why SGA have better DV flow. Need to state (with appropriate references) that patients who are SGA have more often experienced hypoxemia or hypoperfusion in utero, which may predispose them to higher flow through the DV

13.   The authors have spent multiple lines of the discussion addressing change in DV flow with gestational age, and have used this to explain the increased misplacement with older chronologic age. However, the time course is very different than that seen in the chronologic age experienced in this study, which is in the range of hours and not even days. Perhaps the authors should discuss time to DV closure post-natally, rather than gestational age, which is not a variable in this study.

Conclusions: There is no conclusion written for the main body. I do not know if it is needed for this format, as a brief report. If so, then the conclusion from the abstract is adequate.

Author Response

  1. Multiple sentences throughout are missing an article, use inappropriate prepositions, or have commas placed incorrectly. Would benefit throughout from thorough grammatical editing. – this has not been addressed with the first revision.

Revised accordingly.

  1. There remains quite a bit of redundancy in wording. For example:

“Logistic regression model revealed that chronological age positively predicted UVC misalignment, whereas SGA negatively predicted UVC malpositioning” could be shortened with no change in meaning to: Logistic regression modeling identified that chronologic age positively predicted, and SGA negatively predicted, UVC misalignment”.

This is just one example. Excess word use is present throughout the manuscript.

Revised accordingly.

  1. Multiple sentences are incomplete – e.g. they do not start with an appropriate article. This may be acceptable in abstracts, but not in the body of the manuscript.

Revised accordingly.

Abstract:

  1. Methods: Lines 6-7, by saying that you did a retrospective chart review, you have described your methods of carrying out your methodology. It is arguably more important to state your methodology. In this case, your methodology is retrospective cohort study, where your cohort is children who had a UVC placed followed by x-ray. Your methods are chart review. You could say: “We performed a retrospective cohort study of neonates who underwent umbilical venous catheterization followed by x-ray assessment of positioning using chart review.”

Revised accordingly.

  1. Results: Lines 10-11 the authors begin talking about “the two groups” without ever explaining what the two groups will be or who is being compared. I’d suggest at the beginning of the results, report the number of patients with UV catheters (the cohort) and then also reporting the number (%) that were misaligned, which will make it easy for the reader to identify that this is a subgroup of the cohort, and then it is reasonable to assume that the rest are the other group.

Revised accordingly and clarified in diagram.

  1. Results: Lines 11-13. Reporting the results of a logistic regression would usually involve reporting a quantitative amount of change (95% CI) for each unit change in the relevant predictor variable. There needs to be some numeric demonstration of this in the results, as it is the primary outcome. This will justify statements of positive and negative prediction.

Revised accordingly and reported in table 1.

Background:

  1. I like the figure!

Thanks.

  1. Line 44: Either remove “following the incorrect path”, or change the sentence to “…is to explore potential factors that contribute to the UVC misalignment, with misalignment defined as the catheter following the incorrect path”.

Revised accordingly.

Results:

  1. Line 89: This sentence needs reworded. Suggest: The final cohort included 480 patients; 332 met the definition of good alignment and 148 of misalignment (Diagram 1)

Revised accordingly.

  1. Line 90: Should read “The relationships of chronologic age, small for gestational age (etc)….to UVC malpositioning are shown in Table 1.”

Revised accordingly.

  1. Lines 92-95: This is awkward reporting. Normally would be reported along the lines of the following, which is suggested by Zach Bobbitt from Statology (https://www.statology.org/how-to-report-logistic-regression-results/#:~:text=We%20can%20use%20the%20following,%5D%20and%20%5Bresponse%20variable%5D.):

Logistic regression was used to analyze the relationship between [predictor variable 1], [predictor variable 2], … [predictor variable n] and malposition.

It was found that, holding all other predictor variables constant, the odds of malposition occurring [increased or decreased] by [some percent] (95% CI [Lower Limit, Upper Limit]) for a one -unit increase in chronological age.

It was found that, holding all other predictor variables constant, the odds of malposition occurring [increased or decreased] by [some percent] (95% CI [Lower Limit, Upper Limit]) for a one -unit increase in birthweight.

Revised accordingly.

  1. Line 94: how do you justify that SGA is the overall best predictor?

Revised accordingly.

  1. Line 96-97: I found this confusing. On the one hand, the authors have indicated that increasing chronologic age in hours predicts misalignment. But this sentence says there is no difference in the first few hours versus the 2ndor 3rd days. Needs more consistency.

Clarified and removed accordingly.

Discussion:

  1. Lines 111-113: There is still no connection made between how hypoxemia, which increases ductus venosus (DV) flow, would explain why SGA have better DV flow. Need to state (with appropriate references) that patients who are SGA have more often experienced hypoxemia or hypoperfusion in utero, which may predispose them to higher flow through the DV

Revised accordingly.

  1. The authors have spent multiple lines of the discussion addressing change in DV flow with gestational age, and have used this to explain the increased misplacement with older chronologic age. However, the time course is very different than that seen in the chronologic age experienced in this study, which is in the range of hours and not even days. Perhaps the authors should discuss time to DV closure post-natally, rather than gestational age, which is not a variable in this study.

Revised accordingly.

Conclusions: There is no conclusion written for the main body. I do not know if it is needed for this format, as a brief report. If so, then the conclusion from the abstract is adequate.

Added accordingly.
